# The miR-182-5p/NDRG1 Axis Controls Endometrial Receptivity through the NF-κB/ZEB1/E-Cadherin Pathway

**DOI:** 10.3390/ijms232012303

**Published:** 2022-10-14

**Authors:** Seong-Lan Yu, Yujin Kang, Da-Un Jeong, Dong Chul Lee, Hye Jin Jeon, Tae-Hyun Kim, Sung Ki Lee, Ae Ra Han, Jaeku Kang, Seok-Rae Park

**Affiliations:** 1Priority Research Center, Myunggok Medical Research Institute, College of Medicine, Konyang University, Daejeon 35365, Korea; 2Personalized Genomic Medicine Research Center, Korea Research Institute of Bioscience and Biotechnology, Daejeon 34141, Korea; 3Department of Obstetrics and Gynecology, Konyang University Hospital, Daejeon 35365, Korea; 4I-Dream Clinic, Department of Obstetrics and Gynecology, Mizmedi Hospital, Seoul 07639, Korea; 5Daegu cha Fertility Center, CHA University, Daegu 42469, Korea; 6Department of Pharmacology, College of Medicine, Konyang University, Daejeon 35365, Korea; 7Department of Microbiology, College of Medicine, Konyang University, Daejeon 35365, Korea

**Keywords:** endometrial receptivity, miR-182-5p, NDRG1, NF-κΒ/ZEB1/E-cadherin pathway

## Abstract

Endometrial receptivity is essential for successful pregnancy, and its impairment is a major cause of embryo-implantation failure. MicroRNAs (miRNAs) that regulate epigenetic modifications have been associated with endometrial receptivity. However, the molecular mechanisms whereby miRNAs regulate endometrial receptivity remain unclear. Therefore, we investigated whether miR-182 and its potential targets influence trophoblast cell attachment. miR-182 was expressed at lower levels in the secretory phase than in the proliferative phase of endometrium tissues from fertile donors. However, miR-182 expression was upregulated during the secretory phase in infertile women. Transfecting a synthetic miR-182-5p mimic decreased spheroid attachment of human JAr choriocarcinoma cells and E-cadherin expression (which is important for endometrial receptivity). miR-182-5p also downregulated N-Myc downstream regulated 1 (NDRG1), which was studied further. NDRG1 was upregulated in the secretory phase of the endometrium tissues and induced E-cadherin expression through the nuclear factor-κΒ (NF-κΒ)/zinc finger E-box binding homeobox 1 (ZEB1) signaling pathway. NDRG1-overexpressing or -depleted cells showed altered attachment rates of JAr spheroids. Collectively, our findings indicate that miR-182-5p-mediated NDRG1 downregulation impaired embryo implantation by upregulating the NF-κΒ/ZEB1/E-cadherin pathway. Hence, miR-182-5p is a potential biomarker for negative selection in endometrial receptivity and a therapeutic target for successful embryo implantation.

## 1. Introduction

The endometrium, the innermost lining layer of the uterus, is composed of a basal layer and functional layer, which provides an optimal environment for embryo implantation. In particular, cell–cell communication between the endometrial luminal epithelial cells of the functional layer and trophoblast cells of the blastocyst is important for successful implantation. Adhesion molecules, such as integrins, cadherins, and selectins, are differentially expressed in luminal epithelial cells during the endometrial menstrual cycle and play crucial roles in endometrial receptivity [1,2].

MicroRNAs (miRNAs) are small, single-stranded, non-coding RNAs that induce RNA silencing and the post-transcriptional regulation of gene expression via complementary base pairing [3,4,5]. Variable miRNAs are expressed at different endometrial stages during the menstrual cycle, as well as in several pathological gynecological conditions such as infertility, endometriosis, and preeclampsia [6,7]. Numerous reports have demonstrated a correlation between miRNA expression and endometrial receptivity [8]. miR-182 is a member of the miR-183 cluster, which includes miR-183, -182, and -96 [9]. miR-182 has been described as an oncogenic miRNA that targets multiple genes in pancreatic cancer, glioblastoma, lung cancer, breast cancer, and endometrial cancer. In some cancers, miR-182 promotes tumor metastasis by upregulating epithelial–mesenchymal transition (EMT)-related genes [10,11,12,13,14,15]. We previously reported that miR-182 was downregulated in the mid-secretory phase, between 20 and 24 days of the menstrual cycle, during the window of implantation (WOI) in the endometrium [16].

NDRG1 was expressed at higher levels in the secretory phase of the endometrium than in the proliferative phase [16,17]. Altmäe et al. (2017) suggested that NDRG1 acts as a transcriptomic biomarker for endometrial receptivity [18]. However, the relationship between miRNAs and NDRG1 in endometrial receptivity has not been clearly elucidated.

Here, we focused on the role of the miR-182-5p/NDRG1 axis in regulating endometrial receptivity. We found that NDRG1 was downregulated by miR-182-5p and regulated endometrial receptivity by controlling the NF-κΒ/zinc finger E-box binding homeobox 1 (ZEB1)/E-cadherin signaling pathway in endometrial epithelial cells.

## 2. Results

### 2.1. miR-182-5p Was Associated with Defective Endometrial Receptivity

Previously, we performed next-generation sequencing to identify RNAs and non-coding RNAs related to embryo-implantation receptivity and found that miR-182-5p was more highly expressed in the proliferative phase than in the secretory phase during the endometrial receptivity period [16]. To validate the differential expression of miR-182-5p, we quantified miR-182-5p in the proliferative and secretory phases of normal endometrium tissues and during the secretory phase of infertility (Figure 1A). Consistent with our previous results, miR-182-5p was expressed at higher levels in the proliferative phase than in the secretory phase and was more highly expressed during infertility than in the normal secretory phase. Following this, to investigate the role of miR-182-5p in endometrial receptivity, we used human endometrial epithelial cell lines with different receptivities (receptive: RL95-2; non-receptive: AN3-CA). First, the JAr spheroid attachment served as an in vitro model for embryo implantation for both cell lines. The attachment percentage of JAr spheroids to RL95-2 cells was higher than that of AN3-CA cells, which verified that the RL95-2 cells were receptive endometrial epithelial cell lines, as suggested by Ho et al. [19] (Figure 1B). miR-182-5p was expressed at lower levels in RL95-2 cells than in AN3-CA cells (Figure 1C). Hence, we investigated the relationship between the degree of spheroid attachment and miR-182-5p expression. As shown in Figure 1D, spheroid attachment of JAr cells was attenuated in RL95-2 cells overexpressing miR-182-5p, suggesting that miR-182-5p expression correlated negatively with endometrial receptivity. To further investigate this possibility, we examined the expression pattern of E-cadherin, which plays an important role in cell adhesion. E-cadherin expression was downregulated by miR-182-5p overexpression (Figure 1E,F). These results suggest that miR-182-5p expression was inversely related to endometrial receptivity (which is required for successful embryo implantation).

### 2.2. miR-182-5p Negatively Regulated NDRG1 Expression

Most miRNAs induce phenotypic changes by regulating the expression levels of target genes through post-transcriptional mechanisms [3]. To identify target genes associated with endometrial receptivity, we analyzed seven databases including Cupid, MirAncesTar, miRDB, MiRNATIP, MultiMiTar, RNA22, and TargetScan using the microRNA Data Integration Portal (mirDIP; http://ophid.utoronto.ca/mirDIP/; 5 February 2021). NDRG1 was selected as a target gene for miR-182-5p. Moreover, we previously reported a correlation between miR-182 and NDRG1 in a competing-endogenous RNA network [16]. Therefore, to investigate the association between endometrial receptivity and NDRG1, we evaluated NDRG1 expression in the proliferative and secretory phase of the normal endometrium and in the secretory phase of infertility. The mRNA-expression level of NDRG1 was significantly elevated in the secretory phase of the endometrium, compared to that in the proliferative phase, but did not differ significantly between the secretory phases of normal and infertile endometrium tissues (Figure 2A). However, NDRG1 protein abundance was lower in infertile versus normal endometrium tissues, although the difference was not significant (Figure 2B). In addition, NDRG1 mRNA and protein were upregulated in RL95-2 cells, which are receptive endometrial cell lines (Figure 2C,D). These data demonstrate the potential of NDRG1 as a putative target of miR-182-5p for regulating endometrial receptivity. Transfecting synthetic miR-182-5p mimics downregulated NDRG1 expression at both the mRNA and protein levels (Figure 2E,F). Moreover, NDRG1 mRNA expression was significantly increased by miR-182 knockdown (Figure 2G,H). Therefore, these results strongly suggest that miR-182-5p-mediated NDRG1 downregulation was associated with defective endometrial receptivity.

### 2.3. NDRG1 Expression Was Associated with Embryo Implantation

We performed in vitro implantation assays to investigate the correlation between NDRG1 expression and embryo implantation. The spheroid attachment of JAr cells was lower with NDRG1-depleted RL95-2 cells than with control cells (Figure 3A). In contrast, overexpressing NDRG1 in non-receptive AN3-CA cells significantly increased the JAr cell-attachment rate (Figure 3B). These results suggest that NDRG1 plays an important role in successful embryo implantation.

### 2.4. NDRG1 Positively Regulated E-Cadherin Expression in Endometrial Epithelial Cells

E-cadherin plays a critical role in endometrial receptivity, which is important for successful embryo implantation [20,21]. NDRG1 can upregulate E-cadherin expression in pancreatic cancer cells [22]. Therefore, to investigate the relationship between NDRG1 and E-cadherin, which were downregulated by miR-182-5p overexpression, we established an RL95-2 cell line with stably depleted NDRG1 expression using the short-hairpin RNA (shRNA)-mediated gene-silencing method. NDRG1 depletion markedly downregulated E-cadherin mRNA and protein expression (Figure 4A,B). We also established AN3-CA cells that overexpressed NDRG1. NDRG1 overexpression markedly upregulated E-cadherin mRNA and protein expression (Figure 4D,E). These data indicate that NDRG1 positively regulated E-cadherin expression in endometrial epithelial cells. To confirm that NDRG1 can regulate E-cadherin expression, we performed immunofluorescence staining with NDRG1-depleted or -overexpressing cells. E-cadherin expression was attenuated by NDRG1 depletion and enhanced by NDRG1 overexpression (Figure 4C,F). These results suggest that NDRG1 positively regulated E-cadherin expression, which is related to endometrial receptivity and is important for successful embryo implantation.

### 2.5. NDRG1 Downregulated the NF-κΒ/ZEB1 Pathway in Endometrial Epithelial Cells

The NF-κΒ/ZEB1 pathway has been found to repress E-cadherin expression in cancer cell lines [23,24]. Therefore, to investigate the role of NDRG1 in the NF-κΒ/ZEB1 pathway, we analyzed the expression patterns of p65 and ZEB1 in NDRG1-depleted or -overexpressing cells. The mRNA-expression levels of p65 and ZEB1 were significantly elevated by NDRG1 depletion (Figure 5A,B). NDRG1 depletion also induced the protein expression of p65 and ZEB1, suggesting that NDRG1 may regulate endometrial receptivity through the NF-κΒ/ZEB1 pathway (Figure 5C). To test this hypothesis, we examined the expression levels of p65 and ZEB1 in AN3-CA cells overexpressing NDRG1. In contrast to NDRG1 depletion, both p65 and ZEB1 mRNA levels were downregulated by NDRG1 overexpression (Figure 5D,E). NDRG1 overexpression also reduced the protein-expression levels of p65 and ZEB1 (Figure 5F). In addition, the immunofluorescence-staining results agreed with the Western blotting results (Figure 5G–K). These findings suggest that NDRG1 may upregulate E-cadherin expression by downregulating the NF-κΒ/ZEB1 pathway in endometrial epithelial cells.

## 3. Discussion

Endometrial receptivity is the first essential requirement for a successful pregnancy, and its disorder is one of the main causes of infertility due to implantation failure [25]. Several miRNAs have been associated with impaired endometrial receptivity. miR-30d, which is increased in the mid-secretory phase of the endometrium, may be associated with human endometrial receptivity [26,27,28]. Increased miR-30d expression can regulate target genes involved in proliferation and hormonal responses [29]. Shi et al. reported 105 differentially expressed miRNAs in the endometrium of patients with repeated implantation failure (RIF) to identify miRNAs related to impaired endometrial receptivity [30]. miR-200c was identified as a therapeutic target for infertility based on its association with impaired uterine receptivity [31]. It has also been reported that miR-543 downregulation is correlated with impaired endometrial receptivity during the WOI [32]. In addition, miR-183-5p and miR-149 were identified as regulators of endometrial receptivity reported previously [33,34]. Therefore, miRNAs are closely associated with endometrial receptivity.

In this study, we generated evidence suggesting that miR-182 negatively regulates endometrial receptivity. miR-182 was expressed at lower levels in the mid-secretory phase of the endometrium than in the proliferative phase and was expressed at higher levels during infertility. Moreover, miR-182overexpression reduced JAr spheroid attachment in our in vitro implantation model (Figure 1A,D). E-cadherin expression was downregulated in RL95-2 cells overexpressing miR-182-5p (Figure 1E). Epithelial adhesion molecules play essential roles in successful embryo implantation. E-cadherin belongs to the cadherin superfamily and functions as a cell–cell adhesion molecule in the epithelium. Reardon et al. [35] demonstrated that cdh1 disruption results in the loss of uterine function and infertility, where embryos could not attach to the uterus in mice. In humans, E-cadherin expression is expressed at lower levels in the endometrial epithelium of infertile patients than in that of fertile women [36,37]. E-cadherin plays an essential role in embryo attachment to the endometrium, a process known as endometrial receptivity. Recently, it was reported that miR-182 overexpression upregulated EMT-related genes, such as N-cadherin, vimentin, and ZEB1, but downregulated E-cadherin expression in prostate cancer cells [38]. Consistent with previous studies, our data showed that miR-182 led to E-cadherin downregulation, indicating that miR-182 negatively regulates endometrial receptivity.

Endometrial receptivity occurs during the mid-secretory phase of the menstrual cycle. In this study, miR-182 was downregulated in the mid-secretory phase of the endometrium, and its target gene, NDRG1, was upregulated when compared to expression observed during the proliferative and secretory phases of infertility (Figure 2A,B). Moreover, NDRG1 expression was expressed at higher levels in highly receptive RL95-2 endometrial cells than in non-receptive AN3-CA endometrial cells (Figure 2C,D). Moreover, NDRG1 overexpression improved JAr spheroid attachment in our in vitro implantation model (Figure 3). Meng et al. [39] demonstrated that NDRG1 expression was decreased in the uteri of aborted mice. These data indicate that NDRG1 may participate in the endometrial receptivity pathway. Previous findings have demonstrated that NDRG1 served a major role in inhibiting tumor metastasis via EMT inhibition. NDRG1 attenuated EMT and modulated E-cadherin expression by inhibiting caveolin-1 protein expression in colorectal cancer [40]. In addition, NDRG1 also upregulated E-cadherin expression by inhibiting the Smad2 pathway in nasopharyngeal cancer cells [41]. There were some limitations to the spheroid attachment experiment in this study. The spheroids derived from choriocarcinoma may not resemble with characteristics of embryo for implantation. In addition, endometrial epithelial and trophoblast cell lines may not indicate similarity with the cellular response in vivo. As mentioned above, NDRG1 positively regulated E-cadherin expression in endometrial epithelial cells (Figure 4). These results suggest that NDRG1 is closely related to E-cadherin-mediated endometrial receptivity, which is required for successful embryo implantation. In previous studies, NDRG1 inhibited the NF-κΒ signaling pathway, which attenuated E-cadherin expression in pancreatic and colorectal cancers [42,43]. E-cadherin expression was also downregulated by EMT-inducing transcription factors such as ZEB1 and ZEB2 [24]. As shown in Figure 5, we found that NDRG1 downregulated p65, which is related to the NF-κΒ pathway in endometrial epithelial cells. ZEB1 expression was also reduced by NDRG1 overexpression. These results indicate that NDRG1 enhanced endometrial receptivity by upregulating E-cadherin expression via inhibition of the NF-κΒ/ZEB1 pathway.

In this study, we observed that miR-182-5p and its target NDRG1 regulated E-cadherin expression, which plays an important role in embryo implantation. The regulatory mechanism of E-cadherin expression was related to the NF-κΒ/ZEB1 pathway (Figure 6). miR-182-5p mimics led to impaired trophoblastic JAr spheroid attachment in our in vitro implantation model. Therefore, our data suggest that miR-182-5p may serve as a negative biomarker for embryo implantation and a therapeutic target for impaired endometrial receptivity.

## 4. Materials and Methods

### 4.1. Collection of Human Endometrial Tissues

Human endometrial tissues were collected from participants in the proliferative (9–11 menstrual cycle days; mcd) and secretory (20–24 mcd) phases at Konyang University Hospital, and infertility samples of the secretory phase (20–22 mcd) were collected from participants at MizMedi Hospital. Infertile patients did not receive medications, including hormone treatments, during the sample collection period. Endometrial sampling was performed using a disposable uterine sampler (Rampipella, RI.MOS, Mirandola, Italy). An experienced gynecological pathologist histologically determined the menstrual stages of the samples using Noyes criteria [44]. This study was approved by the Bioethics Committee of Konyang University Hospital (institutional review board [IRB] file No. KYUH 2018-11-007) and MizMedi Hospital (IRB file No. MMIRB 2018-3). The characteristics of the volunteers’ endometria are indicated in Table 1.

### 4.2. Cell Culture and Transfections

Human endometrial cancer cell lines (AN3-CA and RL95-2) were obtained from the American Type Culture Collection (Manassas, VA, USA). AN3-CA and RL95-2 were maintained in MEM and DMEM/F-12 (Hyclone, Logan, UT, USA) medium supplemented with 10% fetal bovine serum (FBS; Gibco, Waltham, MA, USA) and 1% penicillin-streptomycin (Hyclone, Logan, UT, USA). The cells were grown at 37 °C in a humidified atmosphere containing 5% CO_2_.

The cells were transfected for 72 h with an miR-182-mimic or miR-182-inhibitor using RNAiMAX (Thermo Fisher Scientific, Waltham, MA, USA) according to the manufacturer’s protocol. AN3-CA cells were transfected with a pcDNA3.1-control or pcDNA3.1-NDRG1 expression vector using Lipofectamine 3000 (Thermo Fisher Scientific, Waltham, MA, USA), and RL95-2 cells were transduced with lentiviral particles expressing an shRNA against NDRG1 mRNA (Sigma Aldrich, St. Louis, MO, USA). NDRG1-overexpressing AN3-CA or NDRG1-knockdown RL95-2 cells were selected with neomycin or puromycin dihydrochloride after transduction, respectively.

### 4.3. RNA Isolation and Quantitative Reverse-Transcription Polymerase Chain Reaction (Qrt-PCR) Analysis

Total RNA was isolated from cells and endometrial tissues using the TRIzol^®^ reagent (Ambion, Austin, TX, USA; Thermo Fisher Scientific, Waltham, MA, USA), according to the manufacturer’s instructions. To analyze Mrna expression, complementary DNAs (cDNAs) was synthesized from 2 μg (Cells) or 5 μg (tissues) total RNAs using Moloney Murine Leukemia Virus reverse transcriptase (Promega, Madison, WI, USA). qRT-PCR was performed with triplicate samples using iQ SYBR Green Supermix (Bio-Rad Laboratories, Hercules, CA, USA) and a CFX Connect Real-Time PCR Detection System (Bio-Rad Laboratories, Hercules, CA, USA). The primers used for real-time PCR are shown in Appendix A. We also identified single peaks in the melting curve of qPCR. The 2^−ΔΔct^ method was used to calculate the relative mRNA-expression levels, using GAPDH as the internal control.

To measure the relative miR-182-expression levels, cDNAs were synthesized with a reverse transcription miR-182-5p or RNU6B primer and the TaqMan miRNA Reverse Transcription Kit (Thermo Fisher Scientific, Waltham, MA, USA), according to the manufacturer’s instructions. Quantitative miRNA expression was performed with TaqMan Master Mix II and TaqMan miRNA assay primers (Thermo Fisher Scientific, Waltham, MA, USA) using a CFX Connect Real-Time PCR Detection System (Bio-Rad Laboratories, Hercules, CA, USA) according to the manufacturer’s protocols. Relative miRNA-expression levels were calculated using the 2^−ΔΔct^ method using RNU6B as an internal control.

### 4.4. Immunoblot Analysis

The cells were lysed on ice using a radioimmunoprecipitation assay buffer (Jubiotech, Daejeon, Korea) containing protease and phosphatase inhibitors (Roche, Basel, Switzerland). Cell lysates were analyzed using a bicinchoninic acid assay (Thermo Fisher Scientific, Waltham, MA, USA). A 30 or 50 μg quantity of protein was separated using sodium dodecyl sulfate-polyacrylamide gel electrophoresis, and transferred to polyvinylidene difluoride membranes (Millipore, Burlington, MA, USA). The blots were incubated with 5% skim milk (Difco, Detroit, MI, USA) for 2 h at room temperature and then probed overnight at 4 °C with primary antibodies (Cell Signaling Technology, Danvers, MA, USA) against E-cadherin (1:1000 dilution, 3195), p65 (1:1000, 8242), ZEB1 (1:1000, 70512), NDRG1 (1:1000, 5196), or GAPDH (1:3000, 5174). On the following day, the blots were incubated with horseradish peroxidase-conjugated secondary antibodies (Millipore, 1:3000) and detected using an Enhanced Chemiluminescence Kit (Thermo Fisher Scientific, Waltham, MA, USA).

### 4.5. Immunofluorescence Analysis

Immunofluorescence analysis was performed as previously described (Yu et al., 2019). Briefly, AN3-CA cells (transfected with pcDNA or a pcDNA-based NDRG1-overexpression vector) or RL95-2 cells (transduced with a control shRNA or an shRNA against NDRG1) were seeded onto coverslips in plates. On the following day, the cells were fixed with 4% paraformaldehyde (Sigma Aldrich, St. Louis, MO, USA) and permeabilized with 0.3% Triton X-100 (Sigma Aldrich, St. Louis, MO, USA). The cells were then blocked with 1% bovine serum albumin solution in phosphate-buffered saline and probed overnight with primary antibodies against p65, ZEB1, and E-cadherin (all from Cell Signaling Technology, Danvers, MA, USA). Following this, the cells were probed with secondary antibodies conjugated with Alexa Fluor 594 or Alexa Fluor 488 (Invitrogen, Waltham, MA, USA). Nuclei were stained with DAPI (Invitrogen, Waltham, MA, USA). A negative control was used to confirm the specificity of the antibodies used in this study (Appendix A). Images were examined and captured using a confocal microscope (LSM710; Carl Zeiss, Oberkochen, Germany).

### 4.6. In Vitro Assay for JAr Spheroid Implantation

To prepare JAr cell spheroids, JAr cells were seeded into a V-bottom microplate (Greiner Bio-one, Kremsmünster, Austria) and incubated in DMEM containing 10% FBS (Gibco, Waltham, MA, USA) and 1% penicillin-streptomycin (Hyclone, Logan, UT, USA) for 24 h in a humidified atmosphere containing 5% CO_2_. Endometrial cells were cultured in separate wells of a 12-well plate until they reached 80% confluency. The spheroids were harvested and co-cultured on monolayers of AN3-CA cells (for 2 h) or RL95-2 cells (for 1 h) that were treated with the miR-182 mimic, NDRG1 shRNA, NDRG1-overexpression vector, or the respective control. To count the attached spheroids, the microplate was inverted and centrifuged. The attached spheroids were counted under a microscope (Olympus, Center Valley, PA, USA). The percentage of spheroid attachment was calculated as the proportion of attached spheroids after running the sample through an inverted centrifuge to the total number of spheroids. The implantation assay was repeated at least thrice.

### 4.7. Statistical Analyses

All experiments were independently performed thrice, and the data are presented as the mean ± SEM. The results were analyzed using Student’s *t*-test or Mann–Whitney test. The thresholds for statistical significance were set at *p* < 0.05 and *p* < 0.01.

## Figures and Tables

**Figure 1 ijms-23-12303-f001:**
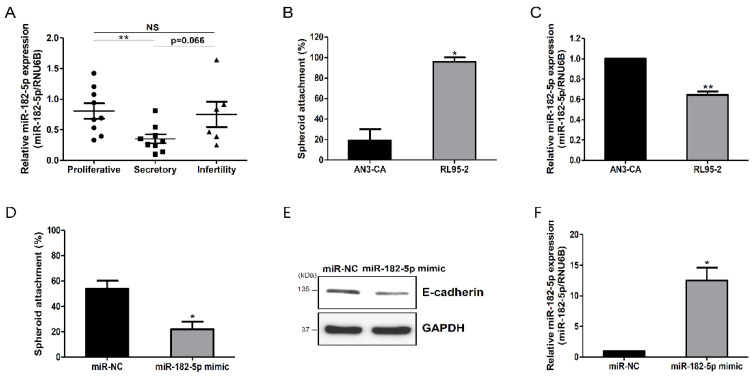
miR-182-5p inhibited the receptive ability of endometrial epithelial cells. (**A**) Analysis of miR-182-5p expression during the proliferative and secretory phases, and the infertile secretory phase of endometrial tissues. (**B**) Assessment of spheroid attachment to non-receptive AN3-CA cells or receptive RL95-2 cells. (**C**) Analysis of miR-182-5p expression in AN3-CA and RL95-2 cells. (**D**) Spheroid-attachment rate of RL95-2 cells transfected with an miR-182-5p mimic. (**E**) Effect of the miR-182-5p mimic on E-cadherin protein expression in RL95-2 cells. (**F**) Confirmation of miR-182-5p expression in RL95-2 cells transfected with the miR-182-5p mimic. The data shown represent the mean ± standard error of the mean (SEM) from three independent experiments (n = 3). * *p* < 0.05; ** *p* < 0.01; NS: not significant.

**Figure 2 ijms-23-12303-f002:**
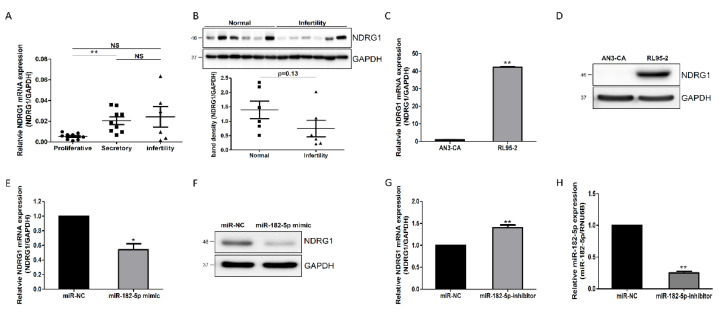
miR-182-5p negatively regulated NDRG1 expression in endometrial epithelial cells. (**A**) Analysis of NDRG1 mRNA expression in proliferative and secretory phases and in secretory phase infertile endometrial tissues. (**B**) Analysis of NDRG1 protein in normal and infertile endometrial tissues. (**C**) Analysis of NDRG1 mRNA expression in non-receptive AN3-CA and receptive RL95-2 cells. (**D**) Analysis of NDRG1 protein expression in AN3-CA and RL95-2 cells. (**E**) Effect of the miR-182-5p mimic on NDRG1 mRNA expression in RL95-2 cells. (**F**) Effect of the miR-182-5p mimic on NDRG1 protein expression in RL95-2 cells. (**G**) Effect of the miR-182-5p inhibitor on NDRG1 mRNA expression in AN3-CA cells. (**H**) Confirmation of miR-182-5p expression in AN3-CA cells transfected with the miR-182-5p inhibitor. The data shown represent the mean ± SEM from three independent experiments (n = 3). * *p* < 0.05, ** *p* < 0.01; NS: not significant.

**Figure 3 ijms-23-12303-f003:**
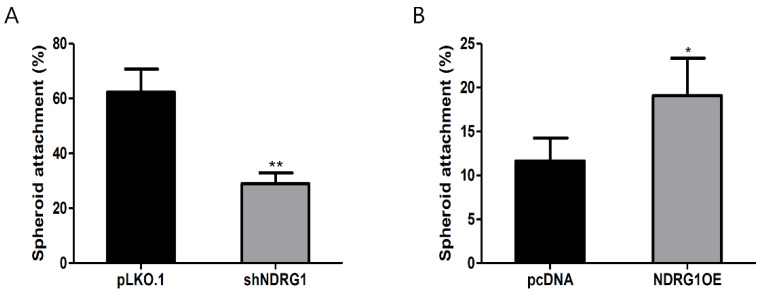
NDRG1 positively affected the attachment of JAr spheroids to endometrial epithelial cells. (**A**) Spheroid-attachment rate to NDRG1-depleted RL95-2 cells. (**B**) Spheroid-attachment rate to NDRG1-overexpressing AN3-CA cells. The data shown represent the mean ± SEM from three independent experiments (n = 3). * *p* < 0.05; ** *p* < 0.01.

**Figure 4 ijms-23-12303-f004:**
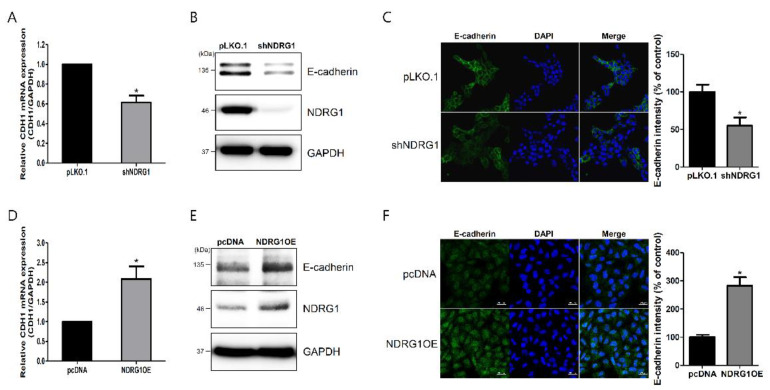
NDRG1 positively regulated E-cadherin expression in endometrial epithelial cells. (**A**) Effect of NDRG1 depletion on CDH1 mRNA expression in RL95-2 cells. (**B**) Effect of NDRG1 depletion on E-cadherin protein expression in RL95-2 cells. (**C**) Immunofluorescence images of E-cadherin expression following NDRG1 depletion in RL95-2 cells. (**D**) Effect of NDRG1 overexpression on CDH1 mRNA expression in AN3-CA cells. (**E**) Effect of NDRG1 overexpression on E-cadherin protein expression in AN3-CA cells. (**F**) Immunofluorescence images of E-cadherin expression induced by NDRG1 overexpression in AN3-CA cells. All images are shown at 400× magnification. Scale bars indicate 20 μm. Nuclei were stained with 4′,6′-diamidino-2-phenylindole dihydrochloride (DAPI). Quantification of immunofluorescence images using image J. The data shown represent the mean ± SEM from three independent experiments (n = 3). * *p* < 0.05.

**Figure 5 ijms-23-12303-f005:**
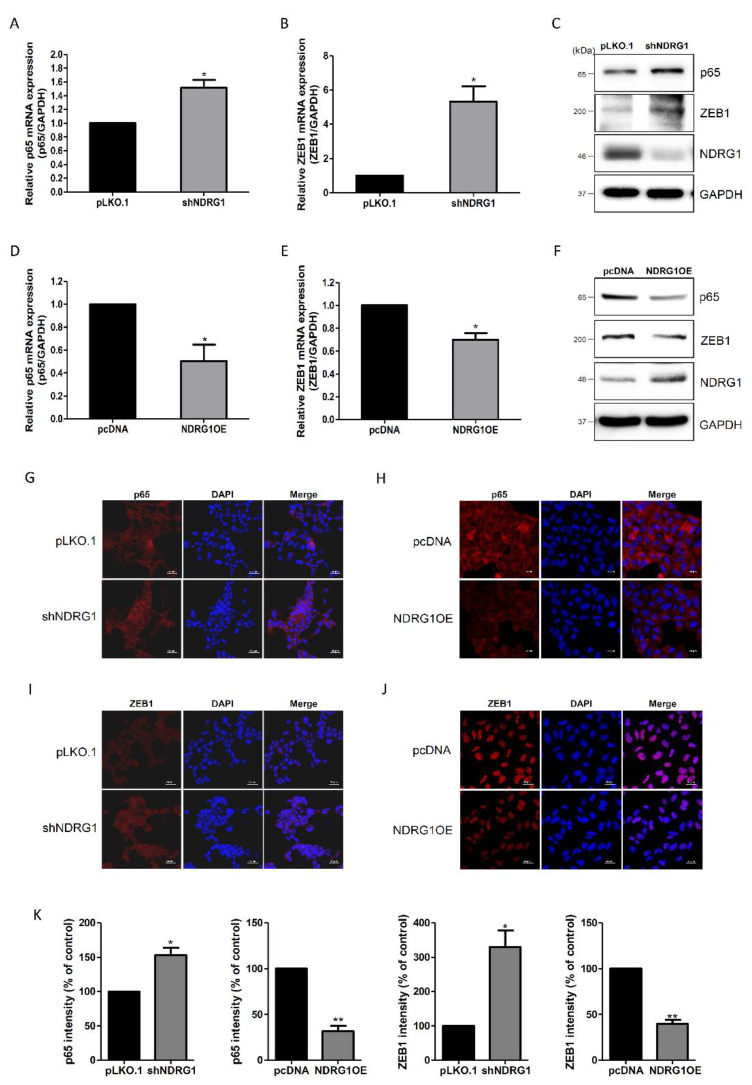
NDRG1 downregulated the NF-κΒ/ZEB1 pathway in endometrial epithelial cells. (**A**) Effect of NDRG1 depletion on p65 mRNA expression in RL95-2 cells. (**B**) Effect of NDRG1 depletion on ZEB1 mRNA expression in RL95-2 cells. (**C**) Analysis of p65 and ZEB1 protein after NDRG1 depletion in RL95-2 cells. (**D**) Effect of NDRG1 overexpression on p65 mRNA expression in AN3-CA cells. (**E**) Effect of NDRG1 overexpression on ZEB1 mRNA expression in AN3-CA cells. (**F**) Analysis of p65 and ZEB1 protein after NDRG1 overexpression in AN3-CA cells. (**G**,**H**) Immunofluorescence images of p65 protein expression in NDRG1-depleted or -overexpressing cells. (**I**,**J**) Immunofluorescence images of ZEB1 protein expression in NDRG1-depleted or -overexpressing cells. (**K**) Quantification of immunofluorescence images using image J. All images are shown at 400× magnification. Scale bars indicate 20 μm. Nuclei were stained with DAPI. The data shown represent the mean ± SEM from three independent experiments (n = 3). * *p* < 0.05; ** *p* < 0.01.

**Figure 6 ijms-23-12303-f006:**
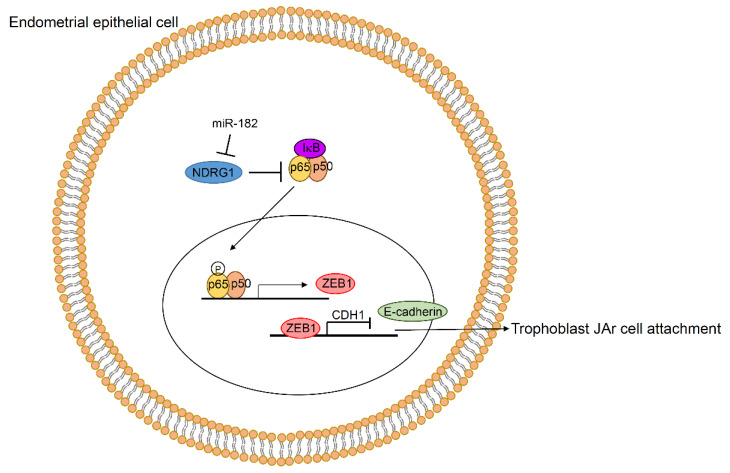
Proposed developmental mechanism for endometrial receptivity disorders, based on miR-182-5p-mediated downregulation of the NDRG1/NF-κΒ/ZEB1/E-cadherin pathway.

**Table 1 ijms-23-12303-t001:** Characteristics of endometrium donors.

Variable/Group	Proliferative Phase (9–11 mcd; n = 9)	Secretory Phase (20–24 mcd; n = 8)	Infertility (20–22 mcd; n = 6)	*p* Value
Age (years)	37.0 ± 3.0	37.8 ± 2.6	38.5 ± 3.7	0.74
BMI (kg/m^2^)	22.8 ± 4.4	22.7 ± 2.8	23.0 ± 4.4	0.99
Number of live births	2.1 ± 0.6	1.9 ± 0.4	0.2 ± 0.4	0.0004
Number of abortions	0.1 ± 0.3	0.4 ± 0.5	0.8 ± 1.3	0.39

The *p* values were determined using an ANOVA analysis. The data shown are presented as the mean ± SD.

## Data Availability

Not applicable.

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
