# Peer review of "The miR-182-5p/NDRG1 Axis Controls Endometrial Receptivity through the NF-κB/ZEB1/E-Cadherin Pathway"

_ijms, 2022, doi:10.3390/ijms232012303_

Round 1
Reviewer 1 Report
The paper describes the roles of miR-182 in endometrial receptivity in an in vitro attachment model.
There should be additional experiments and some critical points.
- A problem with the terminology. Infertility is an ambiguous term; it should be specific.
- In Table 1, the t-test cannot be used to compare three groups!
- In Table 2, I searched the product sizes of the primers used for qPCR in BLAST and found that primers of ZEB1 yield 56 bp amplicon, so please carefully revise the product size for all transcripts and provide the accession numbers of each gene.
- The results lack evidence of direct miR-182 inhibition through using a miR-182 inhibitor.
Author Response
Reviewer #1: The paper describes the roles of miR-182 in endometrial receptivity in an in vitro attachment model.
There should be additional experiments and some critical points.
- A problem with the terminology. Infertility is an ambiguous term; it should be specific.
Response: We thank the reviewer for this comment. Infertility is defined as the inability of a couple to conceive naturally after one year of regular unprotected sexual intercourse (Kamel RM. Reprod Biol Endocrinol. 2010 Mar 6;8:21). We selected infertile patients in accordance with this definition. The diagnosis history is as follows:
Two out of six selected infertile patients experienced two or more repeated miscarriages. Three patients had at least three high-quality embryos implanted into the uterus, but did not become pregnant. For this reason, they were diagnosed with recurrent implantation failure. In addition, one patient was diagnosed with recurrent miscarriages and recurrent implantation failures. To improve clarity, we also modified 'infertility' to 'infertile woman' in the Abstract (page 1, lines 30-31).
- In Table 1, the t-test cannot be used to compare three groups!
Response: We thank the reviewer for the careful and insightful review of our manuscript. We agree that three groups cannot to be used for t-test. Therefore, we used the ANOVA analysis. We apologize for the errors in the manuscript. We have changed the ‘t-test’ to ‘ANOVA test’ in Table 1 of the revised manuscript (page 9, line 297).
- In Table 2, I searched the product sizes of the primers used for qPCR in BLAST and found that primers of ZEB1 yield 56 bp amplicon, so please carefully revise the product size for all transcripts and provide the accession numbers of each gene.
Response: We apologize for the incorrect information on the ZEB1 product size. The product size of ZEB1 was newly revised as shown in the table of Supplementary Materials, and the accession number of each gene was also provided.
- The results lack evidence of direct miR-182 inhibition through using a miR-182 inhibitor.
Response: We appreciate the reviewer’s suggestion. In accordance with the reviewer’s suggestion, we have investigated whether inhibition of miR-182 with a miR-182 inhibitor (Thermo Fisher Scientific, USA) modulates NDRG1 expression in AN3-CA cells. It was confirmed that NDRG1 mRNA expression was increased by inhibition of miR-182. The relevant data was added to Results (page 3, lines 135-136) and in Figures 2G and H of the revised manuscript.

Reviewer 2 Report
The manuscript by Yu et al. expanded on the role of the miR-182-5p/NDRG1 axis in regulating endometrial receptivity. Endometrial samples collected from fertile and infertile women at the proliferative and secretory phases were used, as well as human endometrial epithelial cell lines to link miR-182-5p with receptivity. Authors also propose a mechanism through NDRG1 and the NF-kb/zinc finger E-box binding homeobox 1 77 (ZEB1)/E-cadherin signaling pathway. The model is sound, and the manuscript is well written for the most part. However, I have some concerns that should be addressed. In general, methodology descriptions could be largely improved, as for example, criteria for defining fertile/infertile subjects should be made clearer, since this a critical aspect of the experimental design. I provide below some comments/suggestions that should be addressed before further consideration.
MAJOR:
Lines 90-91, it is unclear how authors defined “receptive: RL95-2; non-receptive: AN3-CA”.
What were the criteria to classify patients as infertile?
Regarding Fig. 1 and elsewhere, how many datapoints were considered for each group (panels B-F). Authors mention that three independent experiments were performed but there is no mention on the final experimental N.
Author should provide more details regarding endometrial tissue collection. For example, lines 278-279, what do the authors mean by “Infertile patients received non-hormone therapy during the sample-collection period”? In the same paragraph, authors should make it clearer whether endometrial sample collection was performed by biopsy. Also, how did the pathologist determined the menstrual stages of the samples? Were the tissues fixed and paraffin-embedded before microscopic evaluation?
Authors should provide more details regarding transfection and RNA isolation methods. For how long were transfections carried out? Was a DNase treatment performed during RNA isolation or before reverse transcription? How much total RNA was used for cDNA synthesis?
How much protein was loaded in the gels before electrophoresis? This information should be provided in the methods.
In Fig. 2, estimated molecular weights of bands detected by western blot should be indicated, ideally by showing the molecular weight marker. Same comment applies to Figs. 4 and 5.
Do authors have any image that represents the spheroid attachment assay? This would be helpful to illustrate how it was evaluated. Image could be added to Fig. 3.
How were percentages of spheroids attachment calculated? More details about methodology should be provided.
In Fig. 4, panel B, why are there two bands for e-cadherin? Which one or both represent the targeted protein? Bands for e-cadherin differ between panels B and E. Authors should clarify on that. Based on immunofluorescence images, e-cadherin abundance does not vary much between shNDRG1-treated and control cells. Are these representative images? Was any sort of semi-quantification performed on immunofluorescence images to arrive to the conclusion that “E-cadherin expression was attenuated by NDRG1 depletion and enhanced by NDRG1 overexpression (Fig. 4C and F)” (Lines 167-168). Same comment applies to Fig. 5.
How was specificity of antibodies used in immunofluorescence staining ensured?
In Fig. 6, authors propose that miR-183 diminishes NDRG1, however, in the legend, it is indicated that “based on miR-182-5p-mediated upregulation of the NDRG1/NF-kb/ZEB1/E-cadherin pathway”. I suggest rephrasing the legend for clarity.
MINOR:
Lines 27-28: “expression was expressed” could be reworded for clarity.
Line 29: “tissues from normal donors” in the abstract should be made clearer. Maybe “from fertile donors”? Correct elsewhere in the manuscript.
Lines 29-30: “during the secretory phase of infertility” sounds awkward. I guess authors meant “during the secretory phase in infertile women”?
Table 1 titles does not match its content.
Table 2 (primer sequences) can be presented as supplemental material.
Line 126: “protein express” should be corrected. Protein abundance would sound better than protein expression.
Scale bar sizes should be mentioned in figure legends since the numbers are hard to be visualized on the figures.
Author Response
Reviewer #2: The manuscript by Yu et al. expanded on the role of the miR-182-5p/NDRG1 axis in regulating endometrial receptivity. Endometrial samples collected from fertile and infertile women at the proliferative and secretory phases were used, as well as human endometrial epithelial cell lines to link miR-182-5p with receptivity. Authors also propose a mechanism through NDRG1 and the NF-kb/zinc finger E-box binding homeobox 1 77 (ZEB1)/E-cadherin signaling pathway. The model is sound, and the manuscript is well written for the most part. However, I have some concerns that should be addressed. In general, methodology descriptions could be largely improved, as for example, criteria for defining fertile/infertile subjects should be made clearer, since this a critical aspect of the experimental design. I provide below some comments/suggestions that should be addressed before further consideration.
MAJOR:
Lines 90-91, it is unclear how authors defined “receptive: RL95-2; non-receptive: AN3-CA”.
Response: Thank you for the valuable comment. Rahnama et al. (2009) proposed two cell lines as a useful model for studying mechanisms involved in human implantation. RL95-2 cells mimic the receptive state of the uterine epithelium, while AN3-CA cells represent the endometrium throughout the nonreceptive phase of the menstrual cycle. Because the two cell characteristics are different, we performed implantation-related experiments in two cell lines. Moreover, two cell lines have been used in a number of papers. We have included references to these papers in the Results.
What were the criteria to classify patients as infertile?
Response: Thank you for raising an important point. Infertility is defined as the inability of a couple to conceive naturally after one year of regular unprotected sexual intercourse (Kamel RM. Reprod Biol Endocrinol. 2010 Mar 6;8:21). We selected infertile patients in accordance with this definition. The diagnosis history is as follows. Two out of six selected infertile patients experienced two or more repeated miscarriages. Three patients had at least three high-quality embryos implanted into the uterus, but did not become pregnant. Therefore, they were diagnosed with recurrent implantation failure. In addition, one patient was diagnosed with recurrent miscarriages and recurrent implantation failures. To improve clarity, we also revised 'infertility' to 'infertile woman' in the Abstract (page 1, line 31).
Regarding Fig. 1 and elsewhere, how many datapoints were considered for each group (panels B-F). Authors mention that three independent experiments were performed but there is no mention on the final experimental N.
Response: We thank the reviewer for this comment. As shown in the results in Figure 1A, we utilized nine proliferative, nine secretory, and six infertile endometrium to perform qRT-PCR. One dot represents one patient. Figures 1B-F were analyzed with a t-test using results derived from three independent experiments as mentioned in the legend. We indicated number of experiments in the Figure legends.
Author should provide more details regarding endometrial tissue collection. For example, lines 278-279, what do the authors mean by “Infertile patients received non-hormone therapy during the sample-collection period”?
Response: We thank the reviewer for this comment. Infertile patients did not receive medications, including hormone treatments, during the sample collection period. Therefore, we have corrected this sentence in Materials and Methods of the revised manuscript (page 8, lines 288-289).
In the same paragraph, authors should make it clearer whether endometrial sample collection was performed by biopsy. Also, how did the pathologist determined the menstrual stages of the samples? Were the tissues fixed and paraffin-embedded before microscopic evaluation?
Response: We thank the reviewer for this comment. The pathologist had determined the menstrual stages based on Noyes criteria (2019) and the endometrial samples were subjected to microscopic evaluation after paraffin embedding. We added the reference in the Materials and Methods of the revised manuscript (page 8 line 292).
Authors should provide more details regarding transfection and RNA isolation methods. For how long were transfections carried out? Was a DNase treatment performed during RNA isolation or before reverse transcription? How much total RNA was used for cDNA synthesis? How much protein was loaded in the gels before electrophoresis? This information should be provided in the methods.
Response: We appreciate the reviewer’s suggestion. In accordance with this suggestion, we marked the time of transfection performed on the cells. We also described the amount of total RNA used for cDNA synthesis and the amount of protein loaded into the gel for electrophoresis in the Materials and Methods of the revised manuscript. Although DNase was not used in the RNA isolation process, DNA contamination is not an issue as the primer sets are comparable in size to products derived from genomic DNA containing introns.
In Fig. 2, estimated molecular weights of bands detected by western blot should be indicated, ideally by showing the molecular weight marker. Same comment applies to Figs. 4 and 5.
Response: We thank the reviewer for this comment. We agree with the reviewer’s suggestion, and have added estimated molecular weights in figures.
Do authors have any image that represents the spheroid attachment assay? This would be helpful to illustrate how it was evaluated. Image could be added to Fig. 3.
Response: We thank the reviewer for this comment. Unfortunately, it was difficult to find spheroid in pictures of the whole area because of average size of spheroid (±120 μm). Hence, we have only part of images to identify spheroid attachment as shown below.
How were percentages of spheroids attachment calculated? More details about methodology should be provided.
Response: We appreciate the reviewer’s suggestion. In accordance with this suggestion, we have added this sentence “The percentage of spheroid attachment was calculated as the proportion of attached spheroids after running the sample through an inverted centrifuge to the total number of spheroids” in Materials and Methods of the revised manuscript (page 10, lines 370–372).
In Fig. 4, panel B, why are there two bands for e-cadherin? Which one or both represent the targeted protein? Bands for e-cadherin differ between panels B and E. Authors should clarify on that.
Response: We thank the reviewer for this comment. In Figure 4A and B, the two bands of E-cadherin corresponded to E-cadherin precursor (upper) and mature form (bottom). The ~120 kDa electrophoretic mobility band is mature E-cadherin produced by E-cadherin processing. As described in figure 4B and E, western blotting was performed to quantify the proteins obtained from RL95-2 and AN3-CA cells, respectively. Therefore, we assume that the band of E-cadherin expresses a different pattern by deviation of expression in each cell type.
Based on immunofluorescence images, e-cadherin abundance does not vary much between shNDRG1-treated and control cells. Are these representative images? Was any sort of semi-quantification performed on immunofluorescence images to arrive to the conclusion that “E-cadherin expression was attenuated by NDRG1 depletion and enhanced by NDRG1 overexpression (Fig. 4C and F)” (Lines 167-168). Same comment applies to Fig. 5.
Response: We thank the reviewer for this comment. We quantified immunofluorescence images using Image J to accurately represent the levels of E-cadherin, p65 and ZEB1 differentially expressed between shNDRG-treated and control cells. We inserted quantified graphs in Figures 4 and 5.
How was specificity of antibodies used in immunofluorescence staining ensured?
Response: We thank the reviewer for this comment. Used antibodies were purchased from Cell Signaling Technology (E-cadherin #3195, p65 #8242 and ZEB1 #70512). These antibodies are classified as capable of performing immunofluorescence staining and referenced in previously published papers.
In Fig. 6, authors propose that miR-182 diminishes NDRG1, however, in the legend, it is indicated that “based on miR-182-5p-mediated upregulation of the NDRG1/NF-kb/ZEB1/E-cadherin pathway”. I suggest rephrasing the legend for clarity.
Response: We thank the reviewer for the careful and insightful review of our manuscript. This sentence is incorrect. Hence we revised this sentence as follows: “based on miR-182-5p-mediated downregulation of the NDRG1/NF-Kb/ZEB1/E-cadherin pathway” in legend of Figure 6 (page 8, line 282).
MINOR:
Lines 27-28: “expression was expressed” could be reworded for clarity.
Response: We have deleted the word “expression”.
Line 29: “tissues from normal donors” in the abstract should be made clearer. Maybe “from fertile donors”? Correct elsewhere in the manuscript.
Response: We thank the reviewer for this comment. We have revised the phrase “from normal donors” to “from fertile donors” (page 1, line 30).
Lines 29-30: “during the secretory phase of infertility” sounds awkward. I guess authors meant “during the secretory phase in infertile women”?
Response: We thank the reviewer for this comment. We have replaced “during the secretory phase of infertility” with “during the secretory phase in infertile women” (page 1, lines 30-31).
Table 1 titles does not match its content.
Response: We thank the reviewer for this comment. We have revised “Characterizing of endometrium samples from donors via quantitative reverse transcription polymerase chain reaction (qRT-PCR) analysis” to “Characteristics of endometrium donors” (page 8, line 297).
Table 2 (primer sequences) can be presented as supplemental material.
Response: We appreciate the reviewer’s suggestion. In accordance with this suggestion, table 2 was included in the table of Supplementary Materials.
Line 126: “protein express” should be corrected. Protein abundance would sound better than protein expression.
Response: We thank the reviewer for this comment. We have revised the phrase “protein express” to “protein abundance” (page 3, line 129).
Scale bar sizes should be mentioned in figure legends since the numbers are hard to be visualized on the figures.
Response: We thank the reviewer for this comment. We have added this sentence “Scale bars indicate 20 μm” in the Figure legends.

Round 2
Reviewer 1 Report
The manuscript has been improved.
Author Response
Reviewer #1: The manuscript has been improved.
Response: We thank the reviewer for his/her thorough and insightful review of our manuscript.

Reviewer 2 Report
Authors did a great job revising the manuscript. A few questions regarding methodologies remain, however.
For western blots, how were molecular weights defined for bands if there is no indication of a marker on the blots?
Regarding antibody specificity for immunostainings, authors state in their response letter that “Used antibodies were purchased from Cell Signaling Technology (E-cadherin #3195, p65 #8242 and ZEB1 #70512). These antibodies are classified as capable of performing immunofluorescence staining and referenced in previously published papers”. This reviewer does not agree that previous literature is sufficient to ensure that one’s actual antibodies work and are specific for one’s samples/protocols. Negative controls should be included in the analysis.
For RNA isolation and qRT-PCR analysis, authors state that they did not perform a DNase digestion step during RNA isolation. RNA samples contaminated with genomic DNA can be problematic in downstream applications. Authors also state that “DNA contamination is not an issue as the primer sets are comparable in size to products derived from genomic DNA containing introns”. This is confusing. The only way to ensure that there is no amplification of gDNA is by using exon-exon spanning primers. Alternatively, single peaks in the melting curve and single bands in electrophoresed PCR products could be used as indicators; however, there is no mention in the manuscript that either verification was performed. Authors should clarify on that.
Author Response
Reviewer #2: Authors did a great job revising the manuscript. A few questions regarding methodologies remain, however.
For western blots, how were molecular weights defined for bands if there is no indication of a marker on the blots?
Response: We sincerely appreciate the reviewer’s suggestion. Western blotting data of Figures 2, 4, and 5 containing size markers are as follows. Data irrelevant to this paper are hidden in boxes. (Please see the attachment)
Regarding antibody specificity for immunostainings, authors state in their response letter that “Used antibodies were purchased from Cell Signaling Technology (E-cadherin #3195, p65 #8242 and ZEB1 #70512). These antibodies are classified as capable of performing immunofluorescence staining and referenced in previously published papers”. This reviewer does not agree that previous literature is sufficient to ensure that one’s actual antibodies work and are specific for one’s samples/protocols. Negative controls should be included in the analysis.
Response: We thank the reviewer for the insightful comment. Images of immunofluorescence staining of E-cadherin, p65, ZEB1 proteins and IgG negative control in AN3-CA cells and RL95-2 cells are as follows. The new data have included in the Supplementary Figure 1. We have added the sentence “A negative control was used to confirm the specificity of the antibodies used in this study (Supplementary Fig. 1).” in Materials and Methods section of the revised manuscript (page 10, lines 360-361). (Please see the attachment)
For RNA isolation and qRT-PCR analysis, authors state that they did not perform a DNase digestion step during RNA isolation. RNA samples contaminated with genomic DNA can be problematic in downstream applications. Authors also state that “DNA contamination is not an issue as the primer sets are comparable in size to products derived from genomic DNA containing introns”. This is confusing. The only way to ensure that there is no amplification of gDNA is by using exon-exon spanning primers. Alternatively, single peaks in the melting curve and single bands in electrophoresed PCR products could be used as indicators; however, there is no mention in the manuscript that either verification was performed. Authors should clarify on that.
Response: We thank the reviewer for carefully reviewing our manuscript. As suggested, the primer sets used in this study were designed by exon-exon spanning primers. Therefore, we have included the sentences “We also identified single peak in the melting curve of qPCR.” in the Materials and Methods (page 9, lines 325-326). The melting curve graph of each gene is as follows. (Please see the attachment)
